# The Effect of a *Bacillus* Probiotic and Essential Oils Compared to an Ionophore on the Rumen Microbiome Composition of Feedlot Cattle

**DOI:** 10.3390/ani13182927

**Published:** 2023-09-15

**Authors:** Dina A. Linde, Dirkjan Schokker, Cornelius J. L. du Toit, Gopika D. Ramkilawon, Este van Marle-Köster

**Affiliations:** 1Department of Animal Science, University of Pretoria, Pretoria 0043, South Africa; 2Wageningen Bioveterinary Research, Wageningen University and Research, 8221 RA Lelystad, The Netherlands; 3Department of Statistics, University of Pretoria, Pretoria 0043, South Africa

**Keywords:** amplicon sequencing, Bonsmara, feed additives, intensive feeding

## Abstract

**Simple Summary:**

Consumer preferences are demanding the removal of antibiotic growth promoters from animal diets. To meet the demand of animal protein for the growing human population, alternative feed additives, such as probiotics and essential oils, need to be investigated to increase the overall efficiency of farm animals. The microorganisms in the rumen are important to the functioning of the animal as they produce the majority of energy the animal uses for production. In this study, the effect of essential oils and a probiotic was compared to the effect of monensin, an antibiotic growth promoter, on the rumen microbiome. There were no substantial differences in the effect of the two natural feed additives compared to monensin on the microorganisms in the rumen. These feed additives hold potential as alternatives to the use of antibiotic growth promoters; large scale production studies are needed to confirm growth performance.

**Abstract:**

The rising concern of antibiotic growth promoter use in livestock has necessitated the investigation into alternative feed additives. The effect of a probiotic and essential oils to an ionophore on the rumen microbiome composition of Bonsmara bulls raised under feedlot conditions was compared. Forty-eight Bonsmara weaners were allocated to four groups: a group with basal diet (CON) and three groups supplemented with monensin (MON), probiotic (PRO), and essential oils (EO). During the 120 days feeding period, rumen content was collected from four animals per group within each phase via a stomach tube for 16S rRNA and internal transcribed spacer (ITS) sequencing as well as volatile fatty acid analysis. In the starter phase, MON had a significantly lower acetate to propionate ratio and a higher *Succinivibrionaceae* abundance. The abundance of *Lachnospiraceae* was significantly higher in EO compared to MON. In the finisher phase, PRO had a significantly higher bacterial diversity. The alpha diversity did not differ between the fungal populations of the groups. The abundance of Proteobacteria was the lowest in PRO compared to the other groups. Limited variation was observed between the rumen microbiome composition of monensin compared to the other treatment groups, indicating that these alternatives can be considered.

## 1. Introduction

The microorganisms in the rumen have been reported to influence the growth and feed efficiency of animals [1,2]. This is due to fermentation of complex carbohydrates by rumen microbiota, which accounts for approximately 70% of the metabolic energy the animal can utilize for maintenance and production [3]. Modification of the rumen microbiota can be seen as a viable strategy to optimize the performance of cattle.

The modification of the rumen microbiome composition through feed additives has been shown to have beneficial effects on the animal’s production and health by reducing excess nitrogen (N) from protein degradation, controlling rumen pH and increasing fibre digestion [4]. Monensin is commonly used in South African feedlot diets as it alters ruminal fermentation and increases feed efficiency [5]. However, due to the development of an antibiotic-resistant bacterium as well as the ban on the usage of antibiotics in subtherapeutic practices by the European Union [6], alternatives that can replace the use of ionophores, such as probiotics or essential oils, need to be investigated.

Probiotics are live microbes that are advantageous to the animal’s health when supplemented in adequate doses [6]. *Bacillus* strains have been used as probiotics and beneficial effects include increased milk quality and growth [7,8]. The inclusion of *Bacillus* in the diet in vitro resulted in the growth of beneficial microorganisms including *Bifidobacterium* and *Lactobacillus* [9]. *Bacillus* bacteria produce a number of antimicrobial compounds that inhibit Gram-positive bacteria and pathogens; however, some also display activity against Gram-negative bacteria [10].

Essential oils (EO) favourably modify rumen fermentation by inhibiting methanogens, and other undesirable microbes, resulting in decreased methane emissions and higher volatile fatty acid (VFA) production [11]. The mode of action of EOs is similar to ionophores in that they interact with the cell membrane, targeting more permeable microorganisms and changing the VFA proportions [12]. The interaction with the cell membrane is influenced by fermentation conditions such as rumen pH and the fermentation substrate [13]. Various EOs can be used; however, synergistic effects have been reported when fed in combinations or blends [12]. Due to EO blends having a similar or superior effect on the animals’ performance [14,15] compared to monensin, EOs hold potential to replace monensin in feedlot diets.

Although there have been studies that showed that EOs can potentially replace monensin, their effect on animal performance and rumen fermentation has been inconsistent [12,16]. There is also limited evidence on the use of probiotics in feedlots. This is the first South African study on the rumen microbiome under intensive feeding making use of feed additives. Understanding the interaction between the rumen microbiota and feed additives can provide a basis on which to develop precision nutrition strategies for optimal production. These strategies may also lead to a decrease in or alternatives to the use of antibiotic growth promoters such a monensin. This study compared the effect of a probiotic and essential oils to an ionophore on the rumen bacterial, archaeal, and fungal populations in South African Bonsmara bulls raised under intensive feedlot conditions.

## 2. Materials and Methods

Ethical approval was received from the University of Pretoria’s Animal Ethical Committee (NAS445/2019) according to the guidelines approved by the veterinary council of South Africa. The trial was completed at the facilities of a commercial feedlot in Edenville, Free State, South Africa (−27.6096553, 27.7221717). Forty-eight Bonsmara weaners (228 ± 22 kg; 10–14 months old) were sourced from the same farm. Natural grazing was used to background the animals for 40 days where after they were randomly divided into four groups: basal diet (CON), the basal diet supplemented with either monensin (MON, 0.3 g/animal/day), probiotic (PRO, 2.75 g/animal/day), or essential oils (EO, 1 g/animal/day). The probiotic consisted of two strains, *Bacillus subtilis* and *Bacillus licheniformis* (3.2 × 10^9^ CFU/g), while the essential oils consisted of eugenol (17%), capsicum (7%), and cinnamaldehyde (11%). All additives were mixed in the feed before being fed to the animals.

The animals were blocked by weight and allocated three to a pen, resulting in twelve animals per group. The animals were fed starter, grower, and finisher diets for 21, 80, and 14 days, respectively. The feed for each phase was mixed at the feed mill on farm, bagged, and marked for the trial. The composition of the diets for each phase was reported in Linde et al. [17]. The animals were processed and received an ear implant (Revalor^®^ S, Intervet GesmbH, Vienna, Austria) as per standard feedlot procedures.

Adaptation of the animals to the starter diet was managed by decreasing the amount of hay supplied while increasing the volume of the starter diet over five days. During the grower and the finisher phases, adaptation of the animals to the new diet occurred over three days by increasing the proportion till fed only the new diet. Water and feed were supplied ad libitum to the animals. Feed intake per pen was calculated by subtracting the refusals of the day from the amount of feed provided the previous day. Feed conversion ratio (FCR) was calculated by dividing the daily feed intake by the average daily gain. The animals were weighed once a week, while rumen content was collected a week before the start of a new phase and slaughter.

From each group, four animals were selected (one per pen) at the start of the trial for collection of rumen content within each phase from the same animal (n = 64). Studies have indicated that four or more samples are sufficient for microbial sequencing [18,19]. A flexible stomach tube was inserted through the animal’s mouth into the ventral sac of the rumen by a registered veterinarian. Samples of rumen microbiome composition collected via stomach tube have been reported to be similar to those collected via cannula if both particles and fluid are obtained [20]. Negative pressure was applied via a dosing gun to draw out rumen content (particles and fluid). To safeguard against saliva contamination, the first 50 mL was removed, and the next 50 mL was instantly frozen in liquid nitrogen and placed in a −80 °C fridge until DNA extraction could be performed.

Fluid from the frozen rumen samples [21] were submitted to the Nutri-lab laboratory of the Department of Animal Science (University of Pretoria) for VFA analysis. For preservation of the rumen fluid samples, orthophosphoric acid (25% H_3_PO_4_) was added and the samples were deposited in a −20 °C freezer until VFA analysis could be performed. Volatile fatty acid concentration was analysed through gas chromatography (SCION GC-456, SCION Instruments, Livingston, Scotland) according to FAO [22] with modifications. The gas chromatograph was fitted with a flame ionization detector, an auto-sampler and CP-WAX 58 (FFAP) CB column with a length of 25 m and a 0.53 mm internal diameter with a 2.0 µm acid-modified chemically bonded polyethylene glycol-film thickness. The oven temperature (100 °C) was sustained for 2 min, then increased to 150 °C, where it was once again sustained for 2 min and then increased to 195 °C. The molar proportions of the VFAs were compared between groups [23].

Thawed rumen content samples (300 mg) were homogenized for twelve minutes at maximum speed (400 × 10 rpm) with a BeadBug homogenizer (Benchmark Scientific, Sayreville, NJ, USA). DNA extraction was completed using a QIAamp PowerFecalPro extraction kit (Cat. No./ID: 51804, Qiagen, Hilden, Germany) following the manufacturer’s guidelines. A Qubit Fluorometer (Invitrogen, Waltham, MA, USA) as well as a Nanodrop ND-1000 Spectrophotometer (Thermo Fisher Scientific, Waltham, MA, USA) were used to determine sample quality. One sample from the probiotic group, collected during the backgrounding phase, was discarded due to low quality. Extracted DNA was sent to Novogene (NovogeneAIT Genomics, Singapore) for 16S rRNA (V3–V4) and ITS1 sequencing using an Illumina NovaSeq 250 (Illumina, San Diego, CA, USA) to generate 250 bp pair-ended raw reads. Average reads per samples generated was 200,126 ± 11,204 for 16S rRNA sequencing and 196,787 ± 16,115 for ITS sequencing. Primers were removed in the data received from NovogeneAIT Singapore. Data were deposited in the NCBI Sequence Read Archive under accession number PRJNA721531.

Both forward and reverse reads were cut at 220 base pairs using DADA2 [24] to enhance the quality of the samples. The Ribosomal Database Project [25] and the UNITE database [26] was used for 16S rRNA and ITS annotation, respectively. Taxonomy was assigned to family level. The data were rarefied, and amplicon sequence variants (ASVs) detected in 5% of the samples less than 10 times were discarded. The alpha diversity of the samples (observed number of ASVs, Shannon diversity, and Chao1 richness indices) were determined using phyloseq [27]. The Shannon index indicates the richness and evenness found within the samples, while the Chao1 index indicates the expected amount of ASVs in the community [28]. Beta diversity was determined with PERMANOVA using the adonis function within vegan v2.5.7 [29] and with a principal coordinate analysis (PCoA) depicting weighted UniFrac distances. One sample from the EO group in the grower phase was identified as an outlier and removed. The Proteobacteria ratio, as an indicator for dysbiosis, was calculated by dividing the Proteobacteria abundance with the combined abundance of Bacteroidetes and Firmicutes [30]. Dysbiosis is indicated by values equivalent or above 0.19.

Significant differences were determined by the Kruskal-Wallis and Dunn tests as well as ANOVA between alpha diversity, relative abundance of the microbes, and the performance traits using statistical packages in R statistical software v4.2.1 [31]. The Holm-Bonferroni procedure was performed for multiple test correction. Significant differences were recognized at *p* < 0.05 and trends were acknowledged at 0.05 < *p* < 0.1.

## 3. Results

### 3.1. Performance Results and VFA Concentrations

Except for daily feed intake, the treatment groups did not differ significantly in performance traits measured (Table 1).

The VFA concentration and proportions differed significantly (*p* < 0.05) across the phases as expected. There was no difference between the VFA concentration of the feed additive groups during the backgrounding period. The acetate and propionate concentrations in the starter phase did differ significantly (*p* < 0.05) between the various treatment groups (Table 2) with MON differing from CON, EO, and PRO.

### 3.2. Alpha and Beta Diversity of the Rumen Microbial Composition

An average count of 116,127 ± 19,264 and 150,668 ± 13,495 reads remained after quality control and chimera removal for the 16S rRNA and ITS sequencing, respectively. From the reads, 41,300 bacterial and archaeal and 35,442 fungal ASVs were identified.

Samples taken during backgrounding, before the feed additives were added to the diets, indicated no significant differences in terms of alpha and beta diversity of the bacterial population. Bacteroidetes was the most abundant phylum during the backgrounding period in the rumen, followed by Firmicutes. *Prevotellaceae* was the most predominant family, followed by *Ruminococcaceae* and *Porphyromonadaceae*.

No significant differences in the bacterial alpha diversity between the feed additive groups were observed in the starter phase (Table 3). The observed number of ASVs and the Chao1 richness index of the bacterial/archaeal population were significantly lower in PRO compared to MON in the grower phase. In the finisher phase, the bacterial/archaeal diversity (Shannon index) between the treatment groups did differ significantly with a higher diversity within PRO compared to CON. The PRO group also had a significantly higher richness (Chao1 index and observed number of ASVs) compared to MON and CON in the finisher phase.

No differences were observed in the fungal population of the rumen of animals in the starter and finisher phases between the feed additive groups (Table 4). In the grower phase, there was a tendency to differ in the richness of the fungal population between CON and PRO.

The principal coordinate analysis (PCoA) (Figure 1) showed that MON did cluster apart from the rest of the treatment groups in both the starter and grower phases, indicating different bacterial compositions. The bacterial composition between the feed additive groups differed significantly in the starter (PERMANOVA, *p* = 0.001) and grower phases (PERMANOVA, *p* = 0.022) as indicated by beta diversity analysis. In the finisher phase, no separate clusters were observed. Although PERMANOVA showed significant differences in terms of the bacterial beta diversity (*p* = 0.006), there was only a tendency to differ between CON and MON (*p* = 0.087) in the finisher phase. The two axes of the PCoA explained 34.3%, 34.7%, and 50.5% of the variance in the bacterial/archaeal composition of the starter, grower, and finisher phases, respectively.

There was no clustering of the treatment groups in terms of the fungal composition within the PCoA (Figure 2) for the starter (PERMANOVA, *p* = 0.125) and grower phases (PERMANOVA, *p* = 0.084). In the finisher phase, the EO group clustered separate from the other treatment groups in the PCoA plot with the beta diversity analysis showing a significant difference in the fungal composition (PERMANOVA, *p* = 0.002). The two axes in the PCoA explained approximately 41.4%, 63.8%, and 61.0% of the microbial composition variation of the treatment groups in the starter, grower, and finisher phases, respectively.

### 3.3. Rumen Microorganism Abundances

Within the starter phase across the treatment groups, the predominant phyla were Bacteroidetes and Firmicutes (Figure 3). The compositional relative abundance and the p-values for the bacterial and archaeal phyla and families were reported in Appendix A. Fibrobacteres differed significantly (*p* = 0.01) across the treatment groups in the starter phase with a lower abundance in MON compared to CON. Within the grower phase, Bacteroidetes was significantly (*p* = 0.045) higher and there was a tendency towards a higher abundance of Euryarchaeota (*p* = 0.069) in MON compared to CON. *Prevotellaceae* and *Succinivibrionaceae* were more abundant within MON compared to CON throughout the whole feedlot period, while *Veillonellaceae* was more abundant within the finisher phase.

In the starter phase, Actinobacteria had a higher abundance (*p* = 0.058), while in the grower phase, Firmicutes had a higher abundance (*p* = 0.045) in MON in comparison with EO. *Succinivibrionaceae* and *Veillonellaceae* were lower in abundance, while *Lachnospiraceae* was higher in abundance in EO compared to MON. There was no difference between MON and EO in the finisher phase.

A difference in the abundance of Actinobacteria (*p* = 0.045) and Fibrobacteres (*p* = 0.005) was observed between MON and PRO within the starter phase with a higher and lower abundance in MON compared to PRO, respectively. Spirochaetes had a tendency towards a difference (*p* = 0.097) with a higher abundance in PRO compared to MON in the starter phase. *Veillonellaceae* and *Succinivibrionaceae* were higher in abundance within MON compared to PRO in the starter phase. Within the grower phase, Fibrobacteres had a significantly higher (*p* = 0.036) abundance in MON compared to PRO. The families *Lachnospiraceae*, *Clostridiales_XI*, *Clostridiales_XIII*, and *Elusimicrobiaceae* were more abundant within PRO compared to CON in the finisher phase.

During the finisher phase, a higher abundance of Proteobacteria was observed in CON, while MON and PRO had a lower abundance (*p* = 0.058). All treatment groups had a Proteobacteria ratio above 0.19, indicating dysbiosis. The Proteobacteria ratio of PRO (0.84 ± 0.14) was significantly lower compared to CON (2.06 ± 0.49) and numerically lower compared to MON (1.21 ± 0.62) and EO (1.59 ± 0.59) in the finisher phase.

Ascomycota was the fungal phylum with the highest abundance across the treatment groups (Figure 4), followed by Neocallimastigomycota. The compositional abundance of the fungal phyla and families can be found in Appendix A. In the grower phase, CON had a tendency towards a lower abundance of Basidiomycota (*p* = 0.056) compared to MON. The abundance of individual phyla did not differ between MON and EO in any of the phases.

A slight difference was observed in the Neocallimastigomycota phylum (*p* = 0.084) between the treatment groups in the starter phase, with the highest abundance in PRO. Ascomycota had a significantly (*p* = 0.029) lower abundance in PRO compared to MON in the grower phase, while a tendency towards a difference in the abundance of Ascomycota (*p* = 0.056) and Basidiomycota (*p* = 0.092) between the treatment groups was observed in the finisher phase.

## 4. Discussion

Studies on natural feed additive alternatives have been inconsistent, with varying results on the production of the animals; an increase, a decrease, or no effect [14]. In this study, the emphasis was on the microbiome composition [18] and a significant difference in production was not expected due to the small sample size. The MON group, however, had a numerically higher LW and ADG compared to the other feed additive groups. In a meta-analysis, monensin decreased feed intake by 3% and increased the feed efficiency by 1% [5]. Other studies have also reported no significant difference in production between MON and EO [14,32]. In the finisher phase, EO had a numerically lower acetate to propionate ratio compared to MON. The inclusion of EOs, such as eugenol [13], in a diet fed more than 91 days resulted in a reduction in acetate concentration [33]. The feed intake of PRO was lower compared to the other treatment groups, which is in contrast to previous studies performed with probiotics, such as *Enterococcus faecium* and *Bacillus*, where DMI was increased [7,34]. Lower intakes can result in slower passage rates that can influence the rumen microbiota [2].

This study compared the effect of essential oils and a probiotic to the effect of an ionophore on the rumen microbiome of Bonsmara cattle under intensive feeding conditions. The rumen microbiota did differ significantly between the phases and was reported in Linde et al. [17]. The bacterial and archaeal microbial composition did exhibit significant differences between the treatment groups within the different phases based on beta diversity (microbiota composition). For the fungal population, the treatment groups tended to differ in the grower phase and differed significantly in the finisher phase. Different growth rates of the various rumen microbes have been reported [35], with anaerobic fungi having a slower growth rate compared to rumen bacteria [36]. Fungal organisms might take longer to adapt or to respond to feed additives.

### 4.1. Monensin vs. Control

Although, numerically, CON had a higher richness and diversity compared to MON, the alpha diversity showed no significant differences. Weimer et al. [37] and Schären et al. [16] observed that monensin supplementation decreased bacterial diversity in the rumen, which has been linked with increased efficiency [38].

Succinate-producing microbes, such as *Succinivibrionaceae* and *Veillonellaceae*, were significantly higher in abundance within MON; both are associated with higher weight gain [1]. The supplementation of monensin is known to impact the fermentation characteristics by decreasing the acetate and butyrate concentration while increasing the propionate concentration [5], resulting in more energy being accessible to the animal. In this study, the MON group had the lowest acetate to propionate ratio within the starter phase in comparison with the other groups. However, in the finisher phase, MON had the highest acetate to propionate ratio. Over the past forty years, research has indicated a decrease in the efficacy of monensin on feed efficiency, which can be partially explained by an increase in dietary energy in feedlots [5] or the adaptation of the rumen microorganisms to monensin [39]. The reduction in the acetate to propionate ratio when using monensin [37] is due to the decrease in the Gram-positive microbes, which are primarily acetate producers, and the likely growth of Gram-negative bacteria, such as succinate producer *Fibrobacter succinogenes*, and *Selenomonas ruminantium*, which converts succinate to propionate.

The abundance of Fibrobacteres was significantly reduced in MON in the starter phase. Fibrobacteres are Gram-negative, obligate anaerobes that are cellulolytic colonisers that produce succinate and acetate [40]. It is, therefore, unexpected that *Fibrobacter* would have a lower abundance within MON compared to CON. Monensin is known to affect Gram-positive bacteria more compared to Gram-negative bacteria [37]. However, recently, a study has observed that monensin can inhibit Gram-negative bacteria as well [41], as observed in this study. Other factors, besides the outer membrane and its presence or absence, determine the vulnerability of bacteria to monensin [37]. The abundance of Fibrobacteres was significantly higher in MON within the grower phase; however, no significant difference between MON and CON was observed in the finisher phase. This may indicate an interaction between the roughage to concentrate ratio, monensin, and Fibrobacteres.

A higher abundance of Euryarchaeota in MON compared to CON was observed throughout the feeding period, with a significant difference in the grower phase. The Euryarchaeota phylum consists mainly of methanogenic archaea, such as *Methanomassilicoccaceae* and *Methanobacteriaceae*, which were observed to be abundant in the MON group. Monensin has been reported to decrease methane emissions by inhibiting bacteria that produce hydrogen resulting in a decrease in the substrates needed for methanogenesis [42], instead of affecting methanogen abundance [16].

Within the grower phase, the Basidiomycota abundance was significantly higher in MON compared to CON. The role of aerobic fungi, such as Basidiomycota, in the rumen is unclear; however, they scavenge for free oxygen within the rumen to ensure an anaerobic environment with Ascomycota [43]. Monensin has been indicated to inhibit anaerobic fungi in the rumen of sheep [44]; as a consequence, the abundance of aerobic fungi might increase. Basidiomycota had a higher abundance within MON in the grower phase, with a higher abundance in CON in the finisher phase. This could be attributed to the interaction between the microbes, the feed additive, and the roughage to concentrate component of the diet.

### 4.2. Monensin vs. Essential Oils

No significant difference between EO and MON was observed in alpha diversity, similar to results reported in dairy [16] and beef [32] cattle, where EO did not alter the diversity in the rumen microbiome. However, Patra and Yu [11] indicated that EO decreased the rumen microbiota diversity. Factors such as ruminant species and age, active component in EO, extraction methods, supplementation period, and dose administered are possible sources of variation on the effect of EOs [33].

Compared to MON, EO was characterised by a low abundance of *Succinivibrionacea* and within the grower phase, a higher abundance of *Lachnospiraceae*. *Lachnospiraceae* is a Gram-positive bacterium and could be an indication of being inhibited by both monensin and EO as they affect more permeable bacteria. The variety of functions executed by *Lachnospiraceae* may affect their relative abundance in digestive tract communities of different hosts [45]. A number of species belonging to the *Lachnospiraceae* family have cellulose-degrading activities and are associated with other cellulolytic microbes. The abundance of *Lachnospiraceae* has been positively correlated with feed efficiency [38] and fermentation in beef cattle [46,47]. In contrast, strains belonging to the family have been found in higher abundance in less efficient animals [48,49,50]. Species of the *Lachnospiraceae* family produce butyrate [51] and a higher butyrate concentration has been reported in more efficient animals [52]. In the finisher phase, a higher butyrate concentration was observed in MON compared to EO; however, EO had a higher butyrate concentration in the starter and grower phases. Various studies [14,32] observed a higher butyrate concentration when the diet was supplemented with EO. Not all species of this family are butyrate producers [51] and further research is required to investigate the correlation between butyrate-producing microorganisms and feed efficiency.

Within the feedlot period, the starter and grower phase had a more observable difference between MON and EO, while there was no significant variation in phylum abundance between MON and EO in the finisher phase. Adaptation of microbes to EOs can occur, which may elucidate the diminishing effects of EO in a feedlot environment over time [53]. The effect of EO on microbial fermentation decreased after six to seven days in a dual flow continuous-culture system [11]. Longer exposure of EO to microbes may result in alterations in the microbiome composition, and the possibility exists that some EOs can be degraded by rumen microbes [54].

In a meta-analysis of the influence of EOs on the rumen microbiome composition, it was observed that the addition of EOs to a diet could lead to a decrease in the eukaryote population [33]. In contrast, this study did not observe any variation in the fungal diversity or phyla abundance between MON and EO in the starter, grower, and finisher phases; this might be due to the similar mode of action between monensin and EO.

### 4.3. Monensin vs. Probiotic

Although the Proteobacteria ratio indicated dysbiosis in all treatment groups within the finisher phase, PRO had a significantly lower Proteobacteria ratio and higher diversity compared to CON. Compared to MON, PRO had a numerically lower Proteobacteria ratio and higher diversity. Cattle are at risk within the finisher phase of a feedlot period as they are fed a diet consisting predominantly of concentrate that can lead to a reduction in pH resulting in dysbiosis in the rumen microbiome [55]. Dysbiosis is characterized by a low diversity in the rumen microbiome [55] and a high Proteobacteria ratio [30]. Probiotics are known to have a stabilizing effect on the rumen microbiome composition and are more effective in stressed animals [56]. Proteobacteria are mostly amylolytic bacteria; however, this phylum does contain many pathogenic bacteria [30]. As dysbiosis interferes with the stability of the microbial community, pathogenic bacteria subsequently take the opportunity to proliferate, resulting in a negative effect on the animal. Such a dysbiosis is well documented in metabolism disorders [55,57]. The supplementation of probiotics is known to influence the diversity, richness, and abundance of microbes, resulting in improved immunity, lower occurrence of metabolic disorders, and increased nutrient digestion and absorption [7].

While Proteobacteria was significantly different between PRO and CON, one of its families, *Succinivibrionaceae*, did not differ between the treatment groups. This family is associated with feed efficiency, as it produces succinate that can be converted to propionate [1]. Spirochaetes was higher in abundance within PRO compared to MON. This is in line with a study where calves were supplemented with *Bacillus subtilis* and *B*. *amyloliquefaciens* [7]. The families *Lachnospiraceae*, *Clostridiales_XIII*, *Clostridiales_XI* and *Elusimicrobiaceae* were more abundant in PRO compared to MON. Hyper-ammonia producing microbes, including *Clostridium sticklandii*, *C. aminophilum*, and *Prevotella ruminicola*, are highly sensitive towards ionophores [58] due in part to their Gram-positive nature.

No *Bacillus* ASVs were identified in this study. This may be due to *Bacillus* not being characterized within the database used or being in such a low abundance that it was not detected by amplicon sequencing. Previous amplicon sequencing-based studies did not detect *Bacillus* species as well [59,60].

Limited literature could be found on the influence of probiotics on the rumen fungal composition. In this study, Neocallimastigomycota tended towards a higher abundance in PRO compared to MON. The Neocallimastigomycota phylum, which consists of anaerobic fungi, has been indicated to be the primary fungal phylum within the rumen [61]; however, Ascomycota was perceived to be the predominant fungal phylum in this study. Ascomycota and Basidiomycota were also indicated to be the predominant phyla in another study that also utilized ITS sequencing [62].

This study investigated the individual effects of essential oils and probiotics in comparison to the effect of monensin on the rumen microbiome. Future studies should consider including a combination of essential oils and probiotics that could work synergistically. A large-scale production trial on the effect of these additives on performance parameters should also be conducted.

## 5. Conclusions

Limited differences were noted in the bacterial, archaeal, and fungal rumen population between the MON group and the other treatment groups, CON, EO, and PRO. The natural feed additives, EO, and PRO, might, therefore, be considered as possible alternatives to the use of MON. However, large-scale production studies will be required for more conclusive evidence. It was also shown that the probiotic group had a higher alpha diversity within the finisher phase, which holds potential as this phase is known to have an increased risk for dysbiosis. A higher diversity is known to be a characteristic of a healthy and resilient rumen microbiome. The effect of MON and EO on the bacterial composition seemed to decrease, whereas the effect of the additives on the fungal population seemed to increase as the feedlot period progressed. Further studies on the adaptation of rumen microbes to diets and dietary components are needed.

## Figures and Tables

**Figure 1 animals-13-02927-f001:**
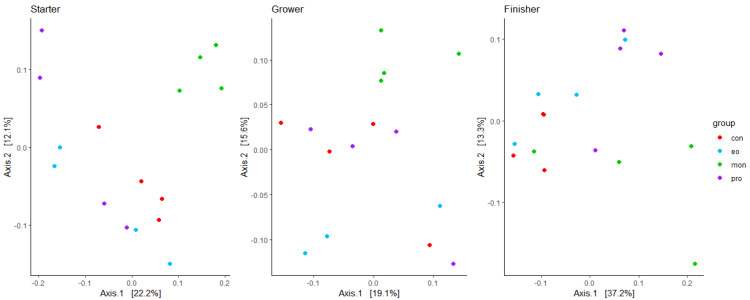
A principal coordinate analysis (PCoA) based on weighted UniFrac distances of the treatment groups in the starter, grower, and finisher phases for the 16S rRNA microbial population. Red depicts the control (CON), blue depicts the essential oils (EO), green the monensin (MON), and purple the probiotic group (PRO).

**Figure 2 animals-13-02927-f002:**
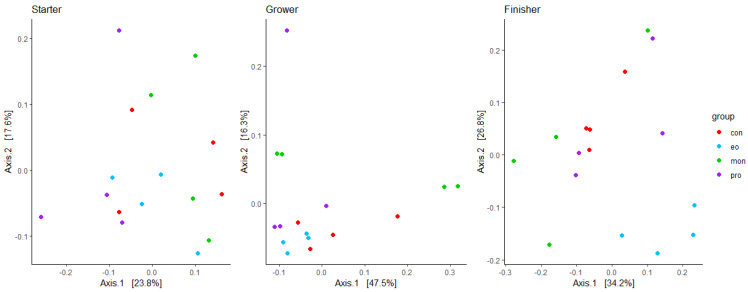
A PCoA plot based on weighted UniFrac distances of the fungal composition of the treatment groups in the starter, grower, and finisher phases. The control group (CON) is depicted in red, essential oils (EO) in blue, monensin (MON) in green, and the probiotic (PRO) group in purple.

**Figure 3 animals-13-02927-f003:**
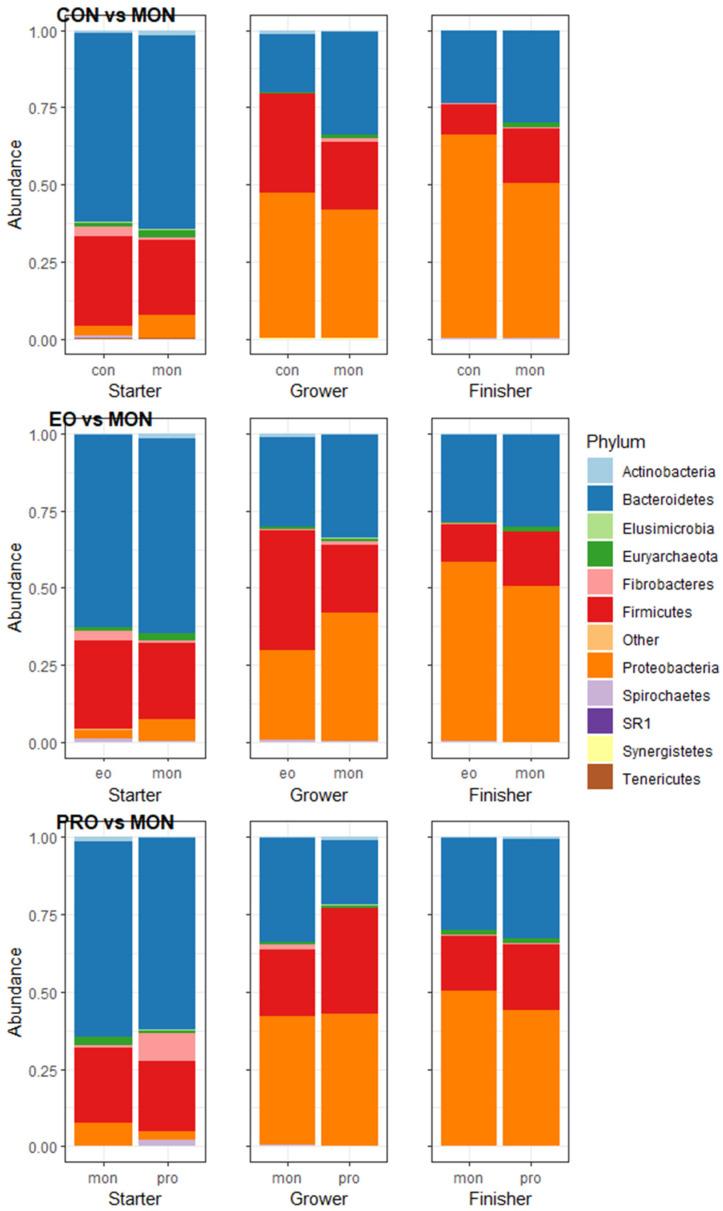
The average relative abundance of the bacterial/archaeal phyla compared between monensin (MON) and control (CON), MON and essential oils (EO), and MON and probiotic (PRO). The x-axis depicts the different samples averaged per treatment group and phase, while the y-axis the compositional relative abundance. Each colour represents a specific phylum as indicated by the legend on the right side of the plot.

**Figure 4 animals-13-02927-f004:**
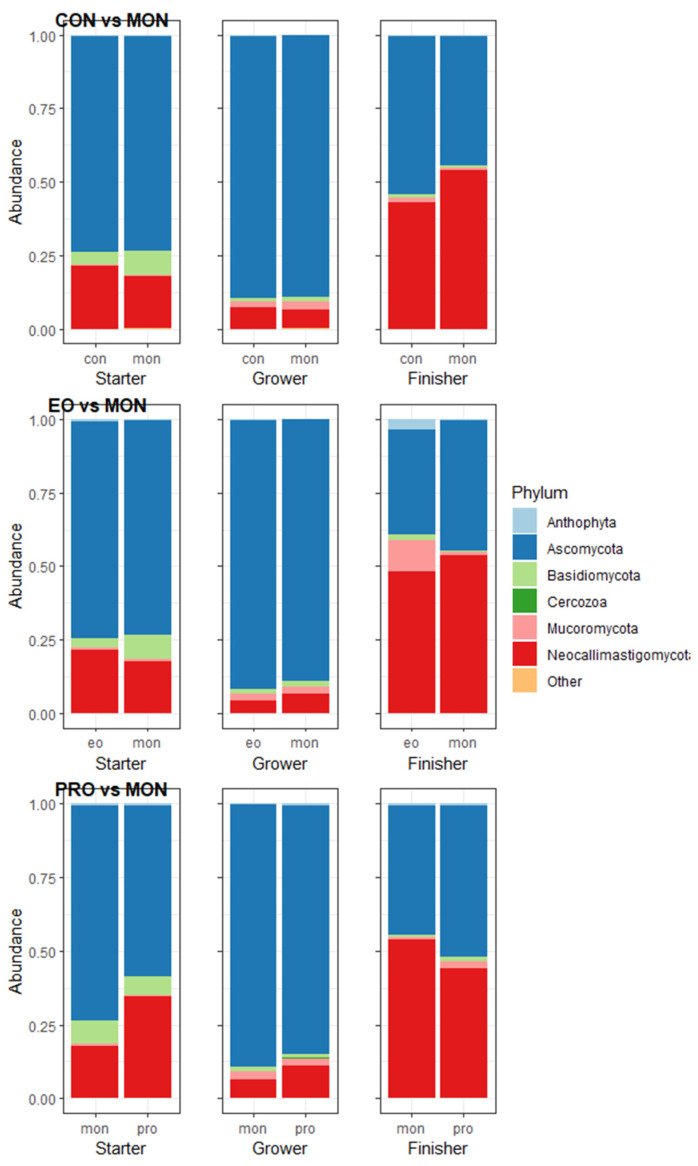
The averaged relative abundance of the fungal phyla compared between monensin (MON) and control (CON), MON and essential oils (EO), and MON and probiotic (PRO). The x-axis depicts the different samples averaged per treatment group and separated by phase, while the y-axis the relative abundance. Each colour represents a specific phylum as indicated by the legend on the right side of the plot.

**Table 1 animals-13-02927-t001:** The average and standard deviation of the live weight (LW), average daily gain (ADG), daily feed intake (DFI), and feed conversion ratio (FCR) for the four treatment groups (48 animals, 12 per group).

Performance Traits	CON	MON	PRO	EO	*p*-Value
LW (kg)	468 ± 26	471 ± 27	455 ± 34	460 ± 30	0.497
ADG (kg/day)	1.81 ± 0.11	1.85 ± 0.08	1.70 ± 0.71	1.68 ± 0.16	0.603
DFI (kg/day)	11.30 ± 0.38 ^a^	11.60 ± 0.71 ^a^	10.50 ± 0.43 ^b^	11.60 ± 0.47 ^a^	0.037 *
FCR	6.25 ± 0.52	6.25 ± 0.46	6.24 ± 0.66	6.91 ± 0.46	0.255

* Significance at *p* < 0.05. ^a,b^ superscripts indicate significant difference within rows at *p* < 0.05.

**Table 2 animals-13-02927-t002:** The average and standard deviation of total volatile fatty acid (tVFA; mmol/L), acetate, propionate, and butyrate (mol/100 mol) concentrations and the acetate to propionate ratio (A:P) of the control (CON), monensin (MON), probiotic (PRO), and essential oils (EO) groups within the three phases.

	CON	MON	PRO	EO	*p*-Value
Starter					
tVFA	70.07 ± 23.17	84.85 ± 18.87	73.03 ± 18.03	81.65 ± 13.27	0.589
Acetate	66.33 ± 2.42 ^a^	57.99 ± 1.54 ^b^	65.26 ± 2.58 ^a^	66.97 ± 2.94 ^a^	0.033 *
Propionate	16.17 ± 4.86 ^a^	28.87 ± 3.38 ^b^	17.67 ± 2.44 ^a^	16.96 ± 2.72 ^a^	0.033 *
Butyrate	12.70 ± 2.38	9.91 ± 1.70	12.73 ± 1.58	12.44 ± 1.96	0.277
A:P	4.52 ± 1.44 ^a^	2.04 ± 0.28 ^b^	3.77 ± 0.61 ^a^	4.08 ± 0.81 ^a^	0.034 *
Grower					
tVFA	105.19 ± 26.02	110.48 ± 13.57	100.01 ± 15.74	91.20 ± 15.15	0.657
Acetate	58.63 ± 0.81	58.46 ± 2.57	57.75 ± 2.41	59.56 ± 2.43	0.724
Propionate	28.19 ± 2.30	26.84 ± 2.57	28.94 ± 3.12	22.43 ± 5.36	0.235
Butyrate	8.19 ± 1.32	9.73 ± 0.29	9.41 ± 0.31	13.16 ± 5.13	0.134
A:P	2.09 ± 0.17	2.21 ± 0.29	2.03 ± 0.31	2.84 ± 0.79	0.474
Finisher					
tVFA	110.59 ± 13.10	94.41 ± 11.23	115.10 ± 18.52	95.03 ± 18.72	0.231
Acetate	56.16 ± 1.07	59.65 ± 3.26	56.23 ± 1.53	55.96 ± 1.56	0.382
Propionate	29.59 ± 1.89	22.52 ± 6.37	29.51 ± 2.11	29.62 ± 1.91	0.531
Butyrate	8.95 ± 1.33	10.96 ± 2.29	9.29 ± 0.99	9.29 ± 1.43	0.562
A:P	1.91 ± 0.15	2.89 ± 0.86	1.92 ± 0.18	1.90 ± 0.16	0.171

* Significance at *p* < 0.05. ^a,b^ superscripts indicate significant difference within rows at *p* < 0.05.

**Table 3 animals-13-02927-t003:** The alpha diversity indices average and standard deviation (observed number of ASVs, Shannon and Chao1 indices) of the bacterial and archaeal population of the control (CON), monensin (MON), essential oils (EO), and probiotic (PRO) treatment groups within the various phases of the feedlot period.

Alpha Diversity Indices	CON	MON	EO	PRO	*p*-Value
Starter					
Observed number of ASVs	1398 ± 48	1213 ± 56	1277 ± 70	1186 ± 105	0.461
Chao1 Index	1402 ± 49	1222 ± 55	1281 ± 70	1196 ± 106	0.492
Shannon Index	6.03 ± 0.15	5.74 ± 0.09	5.88 ± 0.21	5.65 ± 0.21	0.576
Grower					
Observed number of ASVs	805 ± 53 ^ab^	969 ± 37 ^a^	808 ± 47 ^ab^	701 ± 33 ^b^	0.046 *
Chao1 Index	816 ± 54 ^ab^	980 ± 39 ^a^	819 ± 48 ^ab^	708 ± 32 ^b^	0.046 *
Shannon Index	3.92 ± 0.20	4.33 ± 0.12	4.39 ± 0.13	3.95 ± 0.19	0.306
Finisher					
Observed number of ASVs	626 ± 12	608 ± 17	641 ± 66	737 ± 20	0.100
Chao1 Index	629 ± 13	612 ± 17	649 ± 66	742 ± 20	0.108
Shannon Index	2.98 ± 0.13 ^a^	3.63 ± 0.21 ^ab^	3.43 ± 0.24 ^ab^	4.13 ± 0.11 ^b^	0.044 *

* Significance at *p* < 0.05. ^a,b^ superscripts indicate significant difference within rows at *p* < 0.05.

**Table 4 animals-13-02927-t004:** The alpha diversity indices average and standard deviation (observed number of ASVs, Chao1 index, Shannon index) of the fungal population in the control (CON), monensin (MON), essential oils (EO), and probiotic (PRO) treatment groups within the various phases.

Alpha Diversity Indices	CON	MON	EO	PRO	*p*-Value
Starter					
Observed number of ASVs	263 ± 12	275 ± 15	275 ± 10	284 ± 12	0.814
Chao1 Index	264 ± 12	276 ± 15	276 ± 10	285 ± 12	0.782
Shannon Index	3.52 ± 0.17	3.65 ± 0.07	3.69 ± 0.18	3.86 ± 0.12	0.405
Grower					
Observed number of ASVs	304 ± 12 ^a^	287 ± 14 ^ab^	293 ± 10 ^ab^	249 ± 5 ^b^	0.055 **
Chao1 Index	304 ± 12 ^a^	287 ± 13 ^ab^	294 ± 10 ^ab^	250 ± 5 ^b^	0.063 **
Shannon Index	4.10 ± 0.15	3.77 ± 0.31	4.27 ± 0.02	4.17 ± 0.02	0.362
Finisher					
Observed number of ASVs	175 ± 4	151 ± 8	149 ± 5	153 ± 10	0.186
Chao1 Index	175 ± 4	152 ± 9	149 ± 5	153 ± 10	0.183
Shannon Index	3.30 ± 0.05	3.17 ± 0.09	3.21 ± 0.06	3.15 ± 0.08	0.618

** Tendency towards significance at *p* < 0.10. ^a,b^ superscripts indicate significant difference within rows at *p* < 0.05.

## Data Availability

Data were deposited in the Sequence Read Archive of the NCBI with the accession number PRJNA721531.

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
