# Peer review of "The Effect of a Bacillus Probiotic and Essential Oils Compared to an Ionophore on the Rumen Microbiome Composition of Feedlot Cattle"

_animals, 2023, doi:10.3390/ani13182927_

Round 1
Reviewer 1 Report
1. Authors should clearly state this work's main novelty or contribution. Both probiotics and EO have been reported to be beneficial as feed additives in published works, what is the main novelty of this work compared to the literature reference? Also, the small size (3 or 4) of each group is concerning to lead to biased conclusions, making it hard to directly compared with other published work.
2. Since both EO and probiotics have been reported to be advantageous to animals’ growth and modify rumen fermentation with different effects and mechanisms, why do not authors consider including both additives in food?
3. Daily feed intake (DFI) is significantly different within 4 groups, authors need to discuss what effect this difference would have on further analysis.
4. Detailed kit/reagent catalog number needs to be provided in “Materials and Methods” section for better reproducibility.
1. Authors need to give the full name of VFA when first mentioning it.
2. Authors need to put a comma (‘,’) to separate groups of three places on the whole numbers side - for example, “Average reads per samples generated was 200,126….”
Author Response
Dear Reviewer,
Thank you kindly for the comments and suggestions on our manuscript. Please see below our response to your comments and suggestions.
- Comment: authors should clearly state this work's main novelty or contribution. Both probiotics and EO have been reported to be beneficial as feed additives in published works, what is the main novelty of this work compared to the literature reference? Also, the small size (3 or 4) of each group is concerning to lead to biased conclusions, making it hard to directly compared with other published work.
On the novelty or contribution of this work. Monensin is commonly fed in South African feedlot cattle, however, all antibiotic growth promoters are being scrutinized by the South African Department of Agriculture, Forestry and Fisheries and definite bans are expected in the future. Alternatives to its use need to be urgently investigated in a South African context. The aim of this trial was to compare the effects of the probiotic and essential oils on the rumen microbiome to the effect of monensin on the rumen microbiome. Studies have reported that essential oils have similar growth and rumen fermentation results compared to monensin when fed to the animals, however, these results have been inconsistent. We could not find studies where the effect of probiotics was compared to the effect monensin on the rumen microbiome of intensively fed cattle Please see revised lines 69-76.
On the small samples size, studies have shown that four or more samples are sufficient for sequencing studies (Trapnell et al., 2013; Wu et al., 2021), however the samples size is too low to make accurate conclusion for production studies, which is why we do not elaborate on the production results in the discussion. References have been added to materials and methods (lines 110-111).
- Since both EO and probiotics have been reported to be advantageous to animals’ growth and modify rumen fermentation with different effects and mechanisms, why do not authors consider including both additives in food?
It was a consideration in the planning phase of the trial; however, it was decided to first establish the effect each individual feed additive had on the microbiome and then include a combination of probiotics and essential oils in further trials.
- Daily feed intake (DFI) is significantly different between the 4 groups authors need to discuss what effect this difference would have on further analysis.
The sample size is too low for accurate statistical analysis of the production results, which is why we do not elaborate on the significantly different daily feed intake. Lower intakes often result in slower passage rates, which may cause a difference in the microbiota (please see line 327-328). However, as the sample size for the production results is small, it may lead to inaccurate discussions.
- Detailed kit/reagent catalogue number needs to be provided in “Materials and Methods” section for better reproducibility.
The catalogue number has been added to the Material and Methods section (Cat. No. / ID: 51804)
Comments on the Quality of English Language
- Authors need to give the full name of VFA when first mentioning it.
Corrected
- Authors need to put a comma (‘,’) to separate groups of three places on the whole numbers side - for example, “Average reads per samples generated was 200,126….”
Corrected
Thank you for the comments and suggestions. We hope that we have sufficiently answered any concerns you may have had on the manuscript.
Kind Regards,
Dina Linde, Dirkjan Schokker, Lindeque du Toit, Gopika Ramkilawon and Este van Marle-Köster
Reviewer 2 Report
The topic under study is quite new, relevant, practically significant. Previous publications did not reveal the features of changes in the microbiocenosis of the scar sufficiently deeply. The presented manuscript can be a significant step in revealing the mechanisms of changes in microbial communities when the composition of the diet of animals is changed. Of particular interest are the results concerning the effects of essential oils.
Manuscript flaws.
1. Very unfortunate title of the article. I propose to reduce the title by 50%. Perhaps this option is more successful: "Effect of a Bacillus probiotic and essential oils on the rumen microbiome of cattle."
2. Abstract is not informative. Lines 26-31 need to be modified by adding specific results, it is better to add 3-5 digital indicators as well.
3. Key words need to be changed: if the authors were looking for literature on the topic under study, would they really find publications efficiently by entering “Archaea, bacteria, Bonsmara, feed additives, fungi” into the Scopus or Web of Science database? We need to be more careful with reference databases.
4. The introduction contains only 12 references to the literature. The introduction should reveal the state of this most complex scientific problem. At least one paragraph should be added with a detailed analysis of the impact on the microbiome of the studied drugs.
5. Lines 69-70: mention of local ethics committee approval is not enough. Write in more detail what recommendations you followed, describe why your conditions of detention and manipulation cannot be considered inhumane in relation to animals.
6. Lines 75-78: Describe in detail how the drugs are administered to animals. How much and what concentrations, for example, of eugenol, did the animals consume? Very superficially described.
7. Lines 141-143: Statistical processing of data is described incorrectly. It is not clear what the numbers after +- mean (is it (1) error, (2) standard deviation, or (3) standard error)? Have you assessed the normality of the data distribution? For 4 animals, the normality of the samples was most likely absent.
8. Line 152: It is not recommended to round the mean to integers, and the standard deviation to hundredths. In this table, you need to round both in the first row to integers, in the second - fourth - to hundredths.
9. Line 152 and others: Add (mean +- SD, n = 4) in parentheses to the table names.
10. Line 152: Do not use abbreviations in text and tables. This does not greatly reduce the volume of the article, but it makes it much more difficult to understand the results. This applies to both text and tables.
11. Line 152: For the third line of the table, it is necessary to indicate in letters which of the options for the experiment is significantly different from the other options.
12. Line 153: This is not enough. You need to name the comparison method, mention the correction for multiple comparisons.
13. Line 164: In the first line, round up to integers, in the second-fourth - to tenths, in the fifth line - to hundredths.
14. Lines 164, 189 and 201: The reader will be interested to know if the studied characteristics differ significantly for "Starter", "Grower" and "Finisher". In these tables, for each characteristic, a comparison should be made within a column, the results of which should be indicated, for example, in Greek letters or capital letters.
15. Lines 147-167: I don't understand why this part of the results is not divided into one or two subsections, and line 168 is the first subsection of the results? In general, the results section should be more clearly structured into subsections. The discussion should also be subdivided. Lines 293-314 do not have a subheading. It is unacceptable.
16. Line 189: Why are p = 0.100 and p = 0.108 in the penultimate row of the table, while individual samples differ from each other? Something is wrong with the statistical processing of the data.
17. Lines 216 and 230: Instead of three images, I recommend making one. Then it will be clear to the reader which of the studied parameters has a stronger effect. By the location of the points, it will be possible to assume which factor correlates with which of the axes.
18. Lines 247 and 286: The drawing needs to be expanded by 50%, the fonts should be increased by 150%. This is the most informative part of the results. If the authors have more detailed information about the microbiocenosis, it is also desirable to present it in this form in the Results section.
Author Response
Dear Reviewer,
Thank you kindly for the comments and suggestions on our manuscript. Please see below our response to your comments and suggestions.
- Please reduce the length of the title.
The title has been changed to ‘The effect of a Bacillus probiotic and essential oils compared to an ionophore on rumen microbiome composition of feedlot cattle.’ This reduced the title while still emphasising the comparison between the probiotic and essential oils to the ionophore.
- Abstract is not informative. Lines 26-31 need to be modified by adding specific results, it is better to add 3-5 digital indicators as well.
Please see revised lines 28-30 in the abstract.
- Key words need to be changed.
Key words have been changed Amplicon sequencing and intensive feeding has been added (Line 35).
- The introduction contains only 12 references to the literature. The introduction should reveal the state of this most complex scientific problem. At least one paragraph should be added with a detailed analysis of the impact on the microbiome of the studied drugs.
The introduction was modified as suggested. As the impact of the feed additives on the rumen microbiome of South African feedlot cattle are unknown and being investigated, a few studies are mentioned where the impact has been studied to an extent or where the impact of the feed additives on the production of the animals were studied. Please see revised introduction.
- Lines 69-70: mention of local ethics committee approval is not enough. Write in more detail what recommendations you followed, describe why your conditions of detention and manipulation cannot be considered inhumane in relation to animals.
The Animal Ethics Committee of the University of Pretoria uses national guidelines set out by the veterinary council which is recognized by law. The guidelines were adapted from the South African National Standard 10386 ‘The care and use of animals for scientific purposes.’ All procedures were performed by a registered vet.
- Lines 75-78: Describe in detail how the drugs are administered to animals. How much and what concentrations, for example, of eugenol, did the animals consume? Very superficially described.
All the feed additives were mixed into the feed before being fed to the animals. The Bacillus probiotic was 3.2 x 109 CFU/g while the essential oils consisted of 17% eugenol, 11% cinnamaldehyde and 7% capsicum oleoresin. Please see revised lines 88-90.
- Lines 141-143: Statistical processing of data is described incorrectly. It is not clear what the numbers after +- mean (is it (1) error, (2) standard deviation, or (3) standard error)? Have you assessed the normality of the data distribution? For 4 animals, the normality of the samples was most likely absent.
The production data of all 48 animals were used for statistical analysis. The tables show the average of the group followed by the standard deviation. As normality is a standard procedure, it was performed before proceeding with statistical analysis.
- Line 152: It is not recommended to round the mean to integers, and the standard deviation to hundredths. In this table, you need to round both in the first row to integers, in the second - fourth - to hundredths.
Revised. Please see line 168
- Line 152 and others: Add (mean +- SD, n = 4) in parentheses to the table names.
Revised. Please see line 168
- Line 152: Do not use abbreviations in text and tables. This does not greatly reduce the volume of the article, but it makes it much more difficult to understand the results. This applies to both text and tables.
Corrected. Please see line 163.
- Line 152: For the third line of the table, it is necessary to indicate in letters which of the options for the experiment is significantly different from the other options.
The superscripts have been added. Please see line 168.
- Line 153: This is not enough. You need to name the comparison method, mention the correction for multiple comparisons.
We are not certain as to what comparison method or correction for multiple comparisons should be named. If this is regarding the difference in the VFA concentration between phases, the focus of this article is on the effect of the feed additives within the phases. Table 2 does give an indication to the VFAs between phases; however, the focus is within the phases and therefore the statistics for between the phases was not included.
- Line 164: In the first line, round up to integers, in the second-fourth - to tenths, in the fifth line - to hundredths.
Thank you for the suggestion. We have decided to leave the table as is, as we followed the format of other Animals articles.
- Lines 164, 189 and 201: The reader will be interested to know if the studied characteristics differ significantly for "Starter", "Grower" and "Finisher". In these tables, for each characteristic, a comparison should be made within a column, the results of which should be indicated, for example, in Greek letters or capital letters.
Volatile fatty acid analysis was performed between phases and the phases did differ (p < 0.05). However, the analysis was not included as it was not the focus of the article, and it will not contribute to the discussion.
- Lines 147-167: I don't understand why this part of the results is not divided into one or two subsections, and line 168 is the first subsection of the results? In general, the results section should be more clearly structured into subsections. The discussion should also be subdivided. Lines 293-314 do not have a subheading. It is unacceptable.
The following subsections were added to the results: 3.1 Performance results and VFA concentrations; 3.2 Alpha and beta diversity of the rumen microbial composition and 3.3 Rumen microorganism abundances.
- Line 189: Why are p = 0.100 and p = 0.108 in the penultimate row of the table, while individual samples differ from each other? Something is wrong with the statistical processing of the data.
It is procedure to analyse the statistical difference of individual groups within a treatment, if the treatment shows a tendency (p < 0.100) or is close to a tendency (p = 0.100). It is correct that the treatment itself is not statistically significant, however individual groups within the treatment did show statistical significance. The subscripts were removed to avoid confusion.
- Lines 216 and 230: Instead of three images, I recommend making one. Then it will be clear to the reader which of the studied parameters has a stronger effect. By the location of the points, it will be possible to assume which factor correlates with which of the axes.
When one PCoA was drawn encompassing all the phases, the difference between the phases masked the difference between the feed additive groups. To ensure the visibility of the differences between the feed additive groups, three graphs were drawn. For the difference between the phases, please see Linde et al., 2022.
- Lines 247 and 286: The drawing needs to be expanded by 50%, the fonts should be increased by 150%. This is the most informative part of the results. If the authors have more detailed information about the microbiocenosis, it is also desirable to present it in this form in the Results section.
The graphs have been enlarged.
Thank you for the comments and suggestions. We hope that we have sufficiently answered any concerns you may have had on the manuscript.
Kind Regards,
Dina Linde, Dirkjan Schokker, Lindeque du Toit, Gopika Ramkilawon and Este van Marle-Köster
Reviewer 3 Report
There must be a table that promptly indicates the food formulas of the beginning, growth and completion. The foregoing, since the composition of the diet is essential for the growth and type of microbiome that will develop in each of the phases of initiation, growth and completion. Each start, grow, and finish diet must contain the detailed nutritional and energetic characteristics of each feeding phase.THERE ARE SOME ERRONOUS CALCULATIONS IN TABLE 1. 1. It says 6.25 in food conversion and should say 6.27 in MON treatment. 2. It says 6.25 in feed conversion and should say 6.24 in WITH treatment. 3. It says 6.24 in food conversion and should say 6.18 in PRO treatment. 4. It says 6.91 in feed conversion and should say 6.91 in EO treatment. THE ABOVE INDICATES THAT IF THE EO TREATMENT IS CORRECT, THEN THE OTHER 3 TREATMENTS ARE INCORRECT IN THE FEED CONVERSION CALCULATIONS. Therefore, these calculations must be corrected in the first 3 treatments.
Author Response
Dear Reviewer,
Thank you kindly for the comments and suggestions on our manuscript. Please see below our response to your comments and suggestions.
- There must be a table that promptly indicates the food formulas of the beginning, growth and completion. The foregoing, since the composition of the diet is essential for the growth and type of microbiome that will develop in each of the phases of initiation, growth, and completion. Each start, grow, and finish diet must contain the detailed nutritional and energetic characteristics of each feeding phase.
Diet composition was reported in Linde et al., 2022. Please refer to lines 94-95.
- THERE ARE SOME ERRONOUS CALCULATIONS IN TABLE 1. 1. It says 6.25 in food conversion and should say 6.27 in MON treatment. 2. It says 6.25 in feed conversion and should say 6.24 in WITH treatment. 3. It says 6.24 in food conversion and should say 6.18 in PRO treatment. 4. It says 6.91 in feed conversion and should say 6.91 in EO treatment. THE ABOVE INDICATES THAT IF THE EO TREATMENT IS CORRECT, THEN THE OTHER 3 TREATMENTS ARE INCORRECT IN THE FEED CONVERSION CALCULATIONS. Therefore, these calculations must be corrected in the first 3 treatments.
The feed conversion ratio (FCR) reported in the table was calculated by averaging the feed conversion of all the animals (48) within a group. If the average daily gain average of the group is divided by the average daily feed intake of the group, then the feed conversion ratio suggested in the comment is correct. We feel that it is more accurate to report the average FCR of the animals within the group instead of calculating the FCR based on the averages of the group.
Thank you for the comments and suggestions. We hope that we have sufficiently answered any concerns you may have had on the manuscript.
Kind Regards,
Dina Linde, Dirkjan Schokker, Lindeque du Toit, Gopika Ramkilawon and Este van Marle-Köster
Author Response
Dear Reviewer,
Thank you kindly for the comments and suggestions on our manuscript. Please see below our response to your comments and suggestions.
- Line 93, samples were collected at four stages, and the four cows in each group and stage were the same? Are there any basis for the selection of these four cows?
Four bulls from each group (one per pen) were randomly selected at the start of the trial for rumen content collection via stomach tube. The same bulls were used in each phase for rumen content collection. Please see lines 106-108.
- Line 95, how to determined if the rumen tube was inserted to the ventral sac of the rumen.
A registered veterinarian performed the stomach tubing of the animals. As veterinarians do it on a regular basis, we are confident that the samples collected are representative samples.
- Line 107, How were rumen fluid samples pre-treated for VFA analysis.
Rumen fluid samples were treated with orthophosphoric acid prior to analysis. Please see lines 117-119.
- Line 116, the reagents and instruments in the material method should be marked with the corresponding item number and model as far as possible.
Revised.
- As described in the method, it can be seen that the experiment is divided into four stages. Why is the data of the first stage (the backgrounding period) not presented in the results.
The data from backgrounding was not presented as the feed additives were only added after the animals entered the feedlot (from the starter phase). The backgrounding of the animals was only mentioned to indicate that they grazed the same veldt (field) before entering the feedlot.
- Line 151, please indicate how FCR is calculated.
The feed conversion ratio was calculated by averaging the feed conversion ratio of all 48 animals (12 per group). The daily feed intake was divided by the average daily gain. Please see lines 102-103.
- From Table 1, we do not see any benefit to production from the addition of monensin, probiotics or plant oils, so what is the point of using them?
The samples size in this trial is too small to accurately simulate the production benefits of the feed additives. Various studies have shown the production benefits of these feed additives.
- Line 207-212, The result descriptions of this part can not be well represented in Figure 1 (only PCoA), so it is suggested that the author add the more intuitive flora difference analysis charts, such as adonis analysis.
The permutational multivariate of analysis of variance (PERMANOVA) was calculated using the adonis function within the vegan package (Please see lines 224-229 and lines 240-243). The use of the adonis function of the vegan package has been added to the Materials and Methods section (Lines 149-150).
- Figure 3 and Figure 4, why didn't the four groups be compared together and then divided into three stages for presentation, which I think is conducive to comparison and consistent with the previous results.
The aim of the manuscript is to compare the effect of monensin to the effect of probiotics and essential oils on the rumen microbiome of cattle fed under intensive feeding conditions. The graphs (Figure 3 and 4) were drawn to emphasize the fact that a comparison between monensin and the other feed additives were being investigated.
- This experiment uses a large number of cattle and a large sample size, which is not easy. However, at present, the data presented in the manuscript is not complete, the depth of data mining is not enough, and some analyses are not done, such as LEfSe, PICRUSt2 functional predictions, and spearman correlation analysis.
Spearman’s correlation was performed on the data however the production data was insufficient for correlation analysis. It was therefore decided to exclude the correlation results in fear that the small sample size might bias the results.
- The summary of data in the manuscript should be strengthened, and the presentation mode of the results should be modified to highlight the key points.
Subsections has been added to the increase the presentation of the results: 3.1 Performance results and VFA concentrations, 3.2 Alpha and beta diversity of the microbial composition and 3.3 Rumen microorganism abundances.
- Please reorganize the discussion section, refine the discussion focus, strengthen the logical relationship of the writing, and emphasize and highlight the research significance of the manuscript.
The aim of the article is to compare the effect of monensin, an ionophore, to the effect of probiotics and essential oils on the rumen microbiome composition. The discussion was structured to reflect the aim with sections specifically looking at the comparison of monensin to the control, monensin to essential oils and monensin to probiotics.
This trial was performaned to investigate the alternatives to monensin as the use of all antibiotic growth promoters will be banned in the future in South Africa. The study found that there were no substantial differences between the rumen microbiome composition of monensin, essential oils or probiotics. Small differences were observed that might aid in precision nutrition strategies.
Thank you for the comments and suggestions. We hope that we have sufficiently answered any concerns you may have had on the manuscript.
Kind Regards,
Dina Linde, Dirkjan Schokker, Lindeque du Toit, Gopika Ramkilawon and Este van Marle-Köster
Round 2
Reviewer 1 Report
Thank the authors for addressing most of my questions. The quality of the revised manuscript is significantly improved. Here are some minor recommendations:
Minor:
1. Keywords: Remove the extra commas
2. Typo: “An average count of 116,127 ± 19,264 and 15,0668 ± 13,495 reads”, should be 150,668
3. As the authors reply to my questions, I recommend extending the discussion about the future plan/next steps in the Discussion section, which will guide other scientists in this field.
Author Response
Dear Reviewer,
We are pleased that you are satisfied with our answers. Thank you for you comments and suggestions.
The extra commas in the keywords have been removed and the typo has been corrected.
A paragraph on future studies have been added to the end of the discussion (Lines 635-639).
Thank you for your constrictive comments.
Kind regards,
Dina Linde, Dirkjan Schokker, Lindeque du Toit, Gopika Ramkilawon and Este van Marle-Köster
Reviewer 2 Report
The authors have corrected all the shortcomings well. The manuscript may be recommended for publication.
Author Response
Dear Reviewer,
We are pleased that you are satisfied with our answers.
Thank you for your constrictive comments on the manuscript.
Kind regards,
Dina Linde, Dirkjan Schokker, Lindeque du Toit, Gopika Ramkilawon and Este van Marle-Köster
Reviewer 4 Report
The manuscript has been greatly improved after careful revision by the author. The experimental data is substantial and the method is elaborated clearly, which makes it easier for readers to understand the content. The author also responded well to the questions raised by the reviewers, and some errors in the manuscript have been well corrected.
Author Response
Dear Reviewer,
We are pleased that you are satisfied with our answers as well as the revision of the manuscript.
Thank you for your constrictive comments.
Kind regards,
Dina Linde, Dirkjan Schokker, Lindeque du Toit, Gopika Ramkilawon and Este van Marle-Köster